# Group antenatal care compared with standard antenatal care for Somali-Swedish women: a historically controlled evaluation of the Hooyo Project

Malin Ahrne [1], Ulrika Byrskog [2], Birgitta Essén,[3] Ewa Andersson,[1] Rhonda Small,[1,4] Erica Schytt [5,6]

¹Department of Women's and Children's Health, Karolinska Institutet, Stockholm, Sweden
²School of Education, Health and Social Studies, Dalarna University, Falun, Sweden
³Women's and Children's Health, Uppsala University, Uppsala, Sweden
⁴School of Nursing and Midwifery, Judith Lumley Centre, La Trobe University, Melbourne, Victoria, Australia
⁵Center for Clinical Research Dalarna, Uppsala University, Falun, Sweden
⁶Department of Health and Caring sciences, Western Norway University of Applied Sciences, Bergen, Norway

**Correspondence to**
Malin Ahrne; malin.ahrne@ki.se

## ABSTRACT

**Objectives** Comparing language-supported group antenatal care (gANC) and standard antenatal care (sANC) for Somali-born women in Sweden, measuring overall ratings of care and emotional well-being, and testing the feasibility of the outcome measures.

**Design** A quasi-experimental trial with one intervention and one historical control group, nested in an intervention development and feasibility study.

**Setting** Midwifery-led antenatal care clinic in a mid-sized Swedish town.

**Participants** Pregnant Somali-born women (<25 gestational weeks); 64 women in gANC and 81 in sANC.

**Intervention** Language-supported gANC (2017–2019). Participants were offered seven 60-minute group sessions with other Somali-born women led by one to two midwives, in addition to 15–30 min individual appointments with their designated midwife.

**Outcomes** Primary outcomes were women's overall ratings of antenatal care and emotional well-being (Edinburgh Postnatal Depression Scale (EPDS)) in gestational week ≥35 and 2 months post partum. Secondary outcomes were specific care experiences, information received, social support, knowledge of pregnancy danger signs and obstetric outcomes.

**Results** Recruitment and retention of participants were challenging. Of eligible women, 39.3% (n=106) declined to participate. No relevant differences regarding overall ratings of antenatal care between the groups were detected (late pregnancy OR 1.42, 95% CI 0.50 to 4.16 and 6–8 weeks post partum OR 2.71, 95% CI 0.88 to 9.41). The reduction in mean EPDS score was greater in the intervention group when adjusting for differences at baseline (mean difference −1.89; 95% CI −3.73 to −0.07). Women in gANC were happier with received pregnancy and birth information, for example, caesarean section where 94.9% (n=37) believed the information was sufficient compared with 17.5% (n=7) in standard care (p<0.001) in late pregnancy.

**Conclusions** This evaluation suggests potential for language-supported gANC to improve knowledge acquisition among pregnant Somali-born women with residence in Sweden <10 years. An adequately powered randomised trial is needed to evaluate the effectiveness of the intervention.

## STRENGTHS AND LIMITATIONS OF THIS STUDY

⇒ This study was part of an intervention development and feasibility study powered to detect clinically relevant differences in women's overall ratings of antenatal care and the Edinburgh Postnatal Depression Scale (EPDS) but not its impact in relation to other outcomes.

⇒ The study had a participatory approach that integrated the perspectives of midwives and Somali-born women in the intervention development and in the study design.

⇒ The recruitment goal of 63 women in each group was reached so the study had power to detect differences between the groups in emotional well-being measured with the EPDS at baseline, but not overall ratings of care.

⇒ Emotional well-being measured with the EPDS contributes to the evidence for its use, as it is not yet validated for Somali-speaking women.

⇒ Women were recruited from a single site only, and possible selection bias may further limit the representativeness and generalisability of the results; however, the findings may be useful for future power calculations.

**Trial registration number** ClinicalTrials.gov Registry (NCT03879200).

## INTRODUCTION

To improve maternal health in Sweden and globally, there is a need for innovative evidence-based approaches to antenatal care (ANC).[1–3] Migration interacts with other social health determinants contributing to health inequalities.[4 5] With a changing demography of pregnant women in many high-income countries, rethinking how care is provided to migrant women is necessary to improve pregnancy outcomes.[3 6 7] Studies show that many migrant women in high-income countries are at an increased risk of poor pregnancy

outcomes such as higher incidence of comorbidities,[8] pre-eclampsia,[9] preterm birth,[10–12] post-term birth,[13 14] lower use of some obstetrical interventions,[15] higher risk of emergency caesarean birth, postpartum depression and maternal death,[15] and for infants, higher risk of low Apgar score,[11 14] low birth weight,[8 11 14 16 17] small for gestational age (SGA)[16 17] and perinatal mortality.[8 10 16 18]

For Somali-born women, higher rates of complications during pregnancy and birth compared with the general population of migrant-receiving countries have been reported, such as anaemia and severe hyperemesis,[19] fetal distress, perineal laceration, postpartum haemorrhage,[12] higher incidence of caesarean sections, SGA,[19 20] perinatal deaths[21] and stillbirths.[22 23] A lower rate of preterm birth in Somali-born women has been observed,[16 22 24] and a higher incidence of post-term birth.[20 24]

Qualitative studies[25–28] and systematic reviews show that migrant women more often report poor experiences of maternity care, such as communication problems,[3 29–31] lack of familiarity with care systems,[3 29 30] suboptimal care[3 15 30 32] and discrimination.[3 15 29 31] The risk of maternal and reproductive health inequalities and suboptimal care affecting Somali-born women has been described for decades.[23 33–38] Swedish studies show that migrant women,[39] including Somali-born women,[19] commence ANC later, make fewer ANC visits and are less likely to contact obstetric care for decreased fetal movements.[20] Finally, lower attendance in childbirth preparation and parenting classes during pregnancy among migrant women in Sweden and elsewhere has also been reported.[40–42]

New models of maternity care that address migrant women's specific socioeconomic and psychosocial circumstances have been proposed.[3] One such model is group ANC (gANC), which typically incorporates pregnancy check-ups and group sessions for education and social support in a group of women at similar stages of pregnancy.[43 44] Slightly different models of gANC have been developed in various settings.[45–47] gANC has the potential to increase ANC attendance, improve satisfaction with care and pregnancy outcomes such as lower rates of preterm birth, increased breastfeeding rates and reduced risk of depressive symptoms,[48–51] despite some inconsistency in the evidence.[44 47 52] In a Swedish setting, low-risk women in gANC were more satisfied with the information received about labour and birth, and with the midwives' engagement[53]; however, women with inadequate Swedish proficiency were not included. Studies of gANC for migrant women are scarce, but there are some promising examples of small-scale interventions.[7 54 55] None has been conducted in Sweden, where 26% of the population is foreign born.[56]

The aims of this study were to compare language-supported gANC and standard ANC (sANC) for Somali-born women in Sweden in terms of women's overall ratings of care and their emotional well-being in late pregnancy and 2 months post partum, and to evaluate the feasibility of the outcome measures.

## METHODS

### Study design

A single-site, quasi-experimental study with one intervention and one historical control group was conducted between October 2016 and September 2019, as part of the Hooyo Project ('mother' in Somali), a development and feasibility study of gANC for Somali-born women in Sweden. The intervention group received gANC (May 2017–September 2019) and the control group received sANC (October 2016–May 2017). Questionnaires were used at three time points to assess ratings of care, emotional well-being and a number of secondary outcomes. Additional data were retrieved from patient records. The study was registered in ClinicalTrials.gov (NCT03879200).

### Patient and public involvement

The intervention was developed in collaboration with midwives and members of the Somali community; the process is described in detail elsewhere.[57] Initial focus group discussions with Somali women and men informed the intervention design.[25] A study reference group included midwives and members of the Somali community with relevant professional backgrounds and was engaged in all stages of the intervention development. A bicultural research assistant was responsible for recruiting women to the study and conducting the interviews.

### Participants and recruitment

Participants were pregnant (<25 gestational weeks) Somali-born women. Exclusion criteria were severe health conditions (eg, need of specialist obstetric care or a severe mental health condition). At the first ANC appointment, midwives provided initial oral and written information to all eligible women and if agreed, the bilingual research assistant then provided in-depth information, recruited those interested to the study and obtained consent. Information materials were available in Swedish and Somali.

### Setting

Two ANC clinics were involved in developing the model and one clinic implemented the intervention. The implementing clinic is public and located in a mid-sized town in Sweden, with a mixed socioeconomic uptake area, and has 10 midwives.

ANC in Sweden is free of charge with continuity of care throughout pregnancy, and referral to an obstetrician or other specialist when needed. A minimum of eight to nine midwife appointments is recommended,[58] exclusive of ultrasound for pregnancy dating, which is recommended in 18–20 gestational

weeks. Appointments are usually 30 min. A first, early visit focuses on lifestyle factors. The second visit is usually 45 min, including a detailed patient history. Visits 3–9 include health controls of the woman and unborn baby and provision of information, including information about danger signs like vaginal bleeding, leakage of amniotic fluids, decreased fetal movement and symptoms of pre-eclampsia and where to seek healthcare. Language interpretation can be arranged, either by telephone or face to face. ANC clinics offer additional birth preparation in groups and parenting classes. Partners are welcome to attend both individual appointments and classes.

### The Hooyo gANC intervention

The intervention was a combination of gANC and individual check-ups, with language support and integrated childbirth and parenting education. Participants were offered seven 60-minute group sessions together with other women of similar gestational age, led by one to two midwives. Group rules were set by participants, so partners were mostly but not always invited to participate. Directly before or after each group session, a 15-minute individual appointment was scheduled for health check-ups with the designated midwife. Participants started attending gANC in 20–26 gestational weeks. Additional individual appointments were scheduled if needed. Frequency and total number of appointments followed the Swedish national ANC recommendations.[58] The groups were fixed, meaning that women were assigned to a specific group with the same participants and a specific start and end date. Typically, the sessions started with a presentation of a selected topic, following Swedish national guidelines: lifestyle, pregnancy, birth, practical birth preparations, the newborn baby, breastfeeding (and alternatives), parenthood and relationships.[58] The midwives were to be responsive to the interests and concerns of the group, and to encourage questions, dialogue, etc. Language support was provided by a female interpreter in every group session, usually the same interpreter who was also a trained nurse assistant. The interpreter also served as a co-facilitator of the group. Before start-up, the midwives received 1.5 days of training on person-centred care, use of motivational interviewing in groups and group dynamics. A manual was developed. The midwives planned the sessions and chose relevant pedagogical tools like films or anatomical models.

### Control group

The control group received standard, midwifery-led individual care in accordance with Swedish national guidelines[58] as described above.

### Data collection

Three questionnaires with both closed and open-ended questions were developed by the research team in English and translated to Swedish and Somali and are described in detail in the study protocol.[57] Some questions from the Migrant Friendly Maternity Care Questionnaire were included but adapted slightly.[59] The Edinburgh Postnatal Depression Scale (EPDS) was included in all three questionnaires (see below).[60]

A bilingual research assistant recorded women's responses to the questionnaires during face-to-face or telephone interviews; questionnaire one (Q1) in 21–25 gestational weeks, the second questionnaire (Q2) in late pregnancy (<35 gestational weeks) and the third questionnaire (Q3) for the follow-up at 2 months post partum.

Baseline information on women's health, such as age, obstetric history, height, weight, use of tobacco in early pregnancy, diabetes mellitus type 2 (International Classification of Diseases (ICD) code O24.1), haemoglobin and S-ferritin, was retrieved from patient records. Gestational age at first ANC visit is presented in intervals ≤12, 13–20 and 21–25 gestational weeks (after which women were not eligible). Ultrasound for dating of pregnancy is recommended in Sweden in 18–20 gestational weeks,[58] making week 20 a reasonable upper limit for the second interval. The questionnaires included sociodemographic information (ie, language proficiency, level of education, marital status, occupational and migration factors). The entire household monthly disposable income was self-reported in Swedish kronor. Household size is the self-reported number of persons living in the same household.

### Primary outcomes

Women's overall rating of ANC was assessed in late pregnancy and 2 months post partum with the core question 'When thinking about your overall experience of ANC—in general, have you been happy with the care that you have received?' with response alternatives *always, mostly* (happy with care) and *sometimes, rarely* and *never* (not happy with care).

The EPDS is a 10-item scale initially developed to screen for postnatal depression symptoms, which is validated for use in Swedish during pregnancy[61] and has been translated to Somali but not validated.[62] The validated language versions of the EPDS are routinely used for screening of mothers post partum in child health units in Sweden.[63] The 10 items are scored 0–3 according to severity of the self-reported symptoms, with a maximum score of 30. The scale gives an indication of depressive symptoms over the last 7 days. In Sweden, ≥13 points has been validated as an optimal cut-off for detecting depression in pregnant women.[61] The EPDS was used at baseline, in late pregnancy and post partum.

### Secondary outcomes

Women's ratings of different components of ANC in late pregnancy and 2 months post partum were also assessed. A number of questions in this format were asked: 'Have you been happy with…?' with response alternatives *always, mostly, sometimes, rarely* and *never*. Responses were dichotomised in two categories: *always+mostly* and

*sometimes+rarely+never.* Women's ratings of receiving sufficient information about pregnancy, labour and birth as well as social support were assessed (response alternatives *yes* or *no*) in late pregnancy and 2 months post partum. Social support during pregnancy was measured by modified questions from the Pregnancy Risk Assessment Monitoring System.[64] The Cambridge Worry Scale[65] was included in the questionnaires but responses were not analysed because of overlap with other questions related to worries.

Knowledge about danger signs (vaginal bleeding, leakage of amniotic fluids, decreased fetal movement and severe headache) and where to seek healthcare with specific symptoms (general practitioner (GP), labour ward, emergency room, etc) were assessed in Q2, in the question format 'What would you do if you experienced….' with the following response alternatives *wait/ self-care; contact the ANC midwife; GP or nurse; delivery ward or emergency ward.*

Obstetric outcomes included: ANC parameters (number of ANC visits, number of visits to specialist care, referral to an obstetrician, asked about experience of violence, attendance at parent education), health parameters (haemoglobin (lowest value and last value prior to birth), S-ferritin (lowest value), weight gain during pregnancy, gestational diabetes mellitus (ICD-10 code O24.4)), birth outcomes (induction of labour, oxytocin for dystocia, pain relief, mode of birth, perineal injury, blood loss, breast feeding at the labour ward, length of stay), attendance at postpartum check-up, breastfeeding and body mass index (BMI) at the postpartum visit. The number of ANC visits is presented as the median number of visits, and the proportion of women having 6 or fewer visits, 7–11 visits or more than 11 visits. This was considered more clinically relevant to the Swedish context than using the Adequacy of Prenatal Care Utilization Index as originally planned and stated in the study protocol.[57]

Infant outcomes: gestational age, birth weight, SGA (ICD-10 codes P05.0 and P05.1), large for gestational age (ICD-10 codes P08.0 and P08.1), Apgar score <7 at 5 min, umbilical cord pH (arterial and venous) and neonatal intensive care.

## Sample size calculation

The study was designed to have power to detect clinically relevant differences in women's overall ratings of ANC and a difference in means on the EPDS. The initial sample size calculation of 70 women in each group (with 80% power and an alpha of 20%) was based on a national population study on Swedish-speaking women's satisfaction with ANC 2 months post partum.[66] Our assumption was that the ratings of care would improve, from 65% of women receiving individual care being happy with the ANC care received (*always+mostly* happy with care) to 82% of those receiving gANC.

To have similar power to detect differences in EPDS mean scores, 63 women were required in each group, based on a hypothesised reduction from a mean of 8.0

in the control group to that of 6.0 in the intervention group.[67] To allow for loss to follow-up with 20%, a total of 174 women needed to be recruited.

## Statistical analyses

Women who were recruited to the intervention and control groups and contributed data were analysed according to the intention-to-treat concept.[68] Frequencies and percentages are reported for dichotomous variables; median and interquartile ranges (IQR) are reported for continuous variables. $X^2$ tests were performed to test hypotheses for dichotomous variables, and Mann-Whitney U tests for continuous variables. For the primary outcome 'overall ratings of care', odds ratios (OR) with 95% confidence intervals (CI) were calculated.

An analysis of covariance (ANCOVA) test was performed to test the difference in mean EPDS scores between gANC and standard care at 2 months post partum, adjusted for differences in EPDS at baseline, after the exclusion of women with missing EPDS values (gANC n=41, sANC n=38). P values of <0.05 were deemed statistically significant and all tests were two tailed. All statistical analyses were performed in R version 4.0.1.

## RESULTS

Of 270 eligible women, 145 women were recruited to the study (53.7%): 64 women to gANC and 81 to sANC. The number of women who declined to participate was 106 (39.3% of all eligible women), and 19 women did not meet the inclusion criteria (n=6) or were excluded for other reasons (n=13), such as moving or having a miscarriage.

The Q1 was completed by 129 women (89% of all women recruited to the study) (gANC n=62; sANC n=67); the Q2 by 80 women (55%) (gANC n=40; sANC n=40) and the Q3 by 86 women (59%) (gANC n=44; sANC n=42). Of the women in gANC and sANC, 38 women (59.4%) and 32 women (39.5%) responded to all three questionnaires, respectively.

Baseline characteristics for the sample are described in table 1. The groups were largely similar; however, a larger proportion of women in gANC had less than 6 years of education, spoke Somali well or fluently, had no previous births in Sweden and were involved in home duties, on parental leave or were unemployed. The median length of residence in Sweden in this sample was 7 years.

## Overall ratings of care

Women's overall rating of ANC was assessed through the core question 'When thinking about your overall experience of ANC in general, have you been happy with the care that you have received?' in late pregnancy and 2 months post partum, with the response alternatives *always, mostly, sometimes, rarely* and *never.* The vast majority of women in both groups responded *always* and some responded *mostly.* Very few women (5%), and only in the control group, responded *sometimes,* and there were no responses

**Table 1** Characteristics of the sample

| | Group ANC (n=63) | | Standard ANC (n=73) | |
| --- | --- | --- | --- | --- |
| | n (%) | Median (IQR) | n (%) | Median (IQR) |
| Sociodemographics | | | | |
| Age (years) | | 31 (26.3–33.8) | | 30 (25.0–33.3) |
| Language proficiency, well/fluent | | | | |
| Swedish | 41 (65.1) | | 41 (56.2) | |
| Somali | 62 (98.4) | | 60 (82.2) | |
| English | 15 (23.8) | | 14 (19.2) | |
| Missing | 1 (1.6) | | 8 (11.0) | |
| Completed level of education | | | | |
| <6 years | 32 (50.8) | | 25 (34.2) | |
| 7–9 years | 14 (22.2) | | 18 (24.7) | |
| >10 years | 14 (22.2) | | 13 (17.8) | |
| Missing | 3 (4.8) | | 17 (23.3) | |
| Marital status | | | | |
| Married/engaged/in a relationship | 58 (92.1) | | 66 (90.4) | |
| Divorced/widowed/single | 4 (6.3) | | 0 (0.0) | |
| Missing | 1 (1.6) | | 7 (9.6) | |
| Living with husband or partner | 49 (78.8) | | 55 (75.3) | |
| Missing | 14 (22.2) | | 18 (24.7) | |
| Household size | | 3 (2.0–5.0) | | 4 (3.0–6.0) |
| Occupation | | | | |
| Employed | 12 (19.0) | | 12 (16.4) | |
| Student | 19 (30.2) | | 27 (37.0) | |
| Home duties/parental leave/unemployed | 30 (47.6) | | 23 (31.5) | |
| Missing | 2 (3.2) | | 11 (15.1) | |
| Entire household monthly disposable income (SEK) | | 19 000 (13 000–31 750) | | 22 000 (19 000–28 000) |
| Migration | | | | |
| Length of residence (years) | | 7 (5.0–8.8) | | 7 (4.0–8.0) |
| Reason for migration | | | | |
| Refugee/asylum seeker | 32 (50.8) | | 30 (41.1) | |
| Family ties | 30 (47.6) | | 34 (46.6) | |
| Other | 0 (0.0) | | 1 (1.4) | |
| Missing | 1 (1.6) | | 8 (11.0) | |
| Current migration status | | | | |
| Asylum seeker | 4 (4) | | 1 (1.4) | |
| Permanent residency | 35 (55.6) | | 42 (57.5) | |
| Swedish citizen | 23 (36.5) | | 22 (30.1) | |
| Missing | 1 (1.6) | | 8 (11.0) | |
| Obstetric history | | | | |
| Parity | | | | |
| 0 | 11 (17.5) | | 14 (19.2) | |
| 1–2 | 18 (28.6) | | 18 (24.7) | |
| 3–4 | 18 (28.6) | | 17 (23.3) | |
| ≥5 | 15 (23.8) | | 18 (24.7) | |
| Missing | 1 (1.6) | | 6 (8.1) | |

**Table 1** Continued

| | Group ANC (n=63) | | Standard ANC (n=73) | |
|---|---|---|---|---|
| | n (%) | Median (IQR) | n (%) | Median (IQR) |
| Previous stillbirths/neonatal deaths | 4 (6.5) | | 4 (5.8) | |
| No previous birth in Sweden | 5 (7.9) | | 0 (0.0) | |
| Current pregnancy | | | | |
| Gestational age (weeks) at first ANC visit | | 11 (8.8–14.0) | | 12 (10.0–14.0) |
| ≤12 | 32 (51.6) | | 38 (55.1) | |
| 13–20 | 14 (22.6) | | 21 (30.4) | |
| ≥21–25 (after that not eligible) | 2 (3.2) | | 1 (1.4) | |
| Missing | 14 (22.6) | | 9 (13.0) | |
| Body mass index (kg/m$^2$) at first visit | | 27.8 (24.3–31.4) | | 27.6 (24.7–31.3) |
| Underweight (<18.5) | 3 (4.8) | | 4 (5.8) | |
| Normal weight (18.5–24.9) | 14 (22.6) | | 16 (23.2) | |
| Overweight (25.0–29.9) | 17 (27.4) | | 23 (33.3) | |
| Obesity (≥30) | 20 (32.3) | | 22 (31.9) | |
| Missing | 8 (12.9) | | 4 (5.8) | |
| Tobacco use in early pregnancy | 0 | | 0 | |
| S-ferritin at first visit | | 23.5 (15.8–54.0) | | 23.5 (13.8–42.8) |
| S-haemoglobin at first visit | | 121.0 (113.3–124.8) | | 121.5 (117.3–126.8) |
| Diabetes mellitus type 2 prior to pregnancy | 1 (1.6) | | 1 (1.4) | |

ANC, antenatal care; SEK, Swedish kronor.

in the categories *rarely* and *never* (figure 1). When the variable was dichotomised following the protocol, all women (100%) in the gANC group were happy (*always+mostly*) with the care they had received when asked in late pregnancy, compared with 96% in the control group (sANC). Similarly, 2 months post partum, all (100%) women who participated in gANC were happy with the care they had received, compared with 95% of women in sANC (statistical test not applicable). There were no statistical differences between the groups at either time point when the outcome measure was dichotomised into *always* versus *all*

*other alternatives* (late pregnancy OR 1.42, 95% CI 0.50 to 4.16 and 6–8 weeks post partum OR 2.71, 95% CI 0.88 to 9.41).

### Emotional well-being

At baseline, 61 (98.4%) women in gANC and 64 (95.5%) women in sANC responded to all 10 EPDS questions. In late pregnancy, 39 (97.5%) women in gANC and 37 (92.5%) women in sANC responded, and at the last measurement 2 months post partum, 41 women in both groups, respectively, responded to all items (gANC 93.2%; sANC 97.6%).

Mean EPDS scores were higher in women in gANC than in women in sANC at baseline (gANC 9.19; sANC 5.94; difference in means 3.26; 95% CI 1.56 to 4.96) and in late pregnancy (gANC 8.51; sANC 5.73; difference in means 2.78; 95% CI 0.62 to 4.95) (figure 2). Two months post partum, the mean EPDS score was similar between the groups (gANC 3.90; sANC 5.00; difference in means −1.1; 95% CI −2.80 to 0.61). An ANCOVA test of differences in the reduction of mean values (adjusted for differences at baseline) was made for women who responded to all three questionnaires (gANC n=41; sANC n=38). The reduction in mean EPDS score was greater in the intervention group when adjusting for differences at baseline (mean difference −1.89; 95% CI −3.73 to −0.07).

We also checked the mean EPDS score of women who only responded to Q1. In women who later dropped out of the study, the mean EPDS score at baseline was 7.2

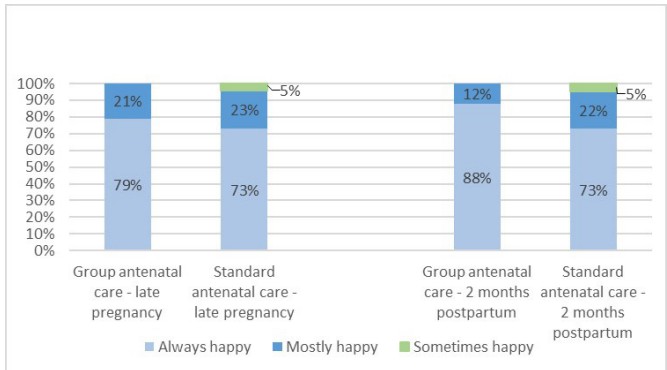

**Figure 1** Women's overall ratings of antenatal care. The proportion of women who were 'always', 'mostly', 'sometimes', 'rarely' and 'never' happy with the care they received in the two different care models, assessed in late pregnancy and 2 months post partum.

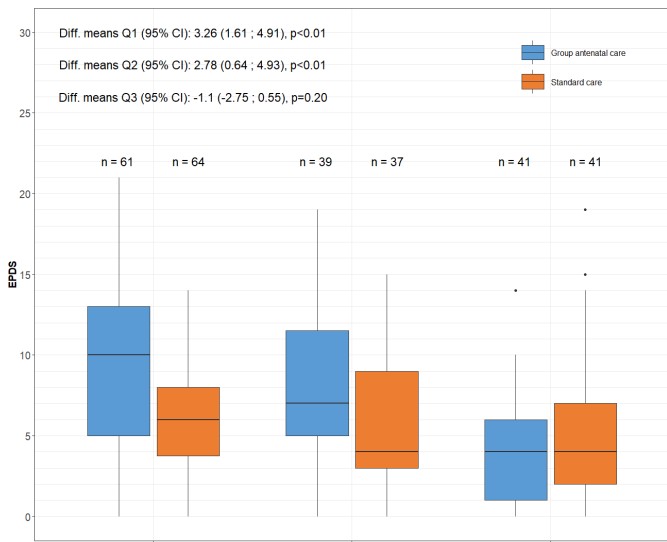

**Figure 2** Median scores on the Edinburgh Postnatal Depression Scale (EPDS). At baseline (Q1), late pregnancy (Q2) and 2 months post partum (Q3), analysed by ANCOVA test of differences. The boxplot also shows minimum and maximum score, and the first and third quartile and the mean differences with 95% CI at each time point. ANCOVA, analysis of covariance.

(median 6.0; IQR 4.0–10.0), which was somewhat lower than the EPDS scores of all women (mean 7.5, median 7.0; IQR 4.0–11.0).

### Secondary outcomes

#### Ratings of care

Women's ratings of specific components of ANC in relation to care model in late pregnancy and 2 months post partum are presented in table 2. Few statistically significant differences could be detected between the groups. Women in gANC were slightly happier with the advice they had received about how to manage common pregnancy disorders and they were more likely to report that the midwives had been encouraging and supportive (table 2). No woman reported experience of negative discrimination by the midwives based on ethnicity, religion, culture, language, etc.

Women's ratings of care during labour and birth as well as during the postpartum stay were also assessed at 2 months post partum, and no differences between the groups were detected. Higher proportions of women in gANC were *always or mostly happy with care* compared with women in sANC, both during labour and birth (gANC 60.3% (n=38); sANC 49.3% (n=36); p=0.423) and on the postpartum ward (gANC 63.5% (n=40); sANC 50.7% (n=37); p=0.218), though neither difference was statistically significant (not presented in table).

#### Sufficient information and social support

Both in late pregnancy and 2 months post partum, women in gANC were generally happier with the information received on different aspects of pregnancy, labour and birth, breast feeding, the care of the newborn baby and physical and emotional changes. Far more women in gANC were happy with information about caesarean section (gANC 94.9%; sANC 17.5%; p<0.001), on the management of female genital mutilation/cutting (FGM/C) (gANC 79.5%; sANC 40.0%; p=0.001) and the father's/partner's role during labour and birth (gANC 94.6%; sANC 42.5%; p<0.001) (table 3) than women in sANC.

Table 3 also shows how women viewed the social support they received. No differences between the groups were observed when assessed in late pregnancy. At 2 months post partum, 97.6% of women who had attended gANC reported that they had someone who could help with a temporary place to live if they should need it, compared with 25.0% of women in sANC (p<0.001). Additionally, 95.2% in gANC reported that they had made new friends through the ANC clinic during pregnancy, compared with 43.9% of women in standard care (p<0.001).

#### Knowledge of danger signs

Knowledge of danger signs and who to contact if experiencing severe symptoms were assessed in late pregnancy (not presented in table). A significantly higher proportion of women in gANC responded that they would contact the delivery ward in case of vaginal bleeding (gANC 54.0%; sANC 28.8%; p=0.004) or leakage of amniotic fluid (gANC 55.6%; sANC 26.0%; p=0.003). For severe headache and changes in vision, 46.0% of women in gANC said they would contact the emergency room compared with 23.3% of women in standard care (p=0.037). Of women in gANC, 30.2% responded that they would contact the delivery ward if experiencing reduced fetal movements, compared with 17.8% of women in sANC (p=0.526).

#### Obstetric outcomes

Obstetric outcomes in relation to care model are described in table 4. Compared with women in standard care, women in gANC made more visits to the antenatal clinic and to specialist care; however, the other health parameters were similar between the groups. Labour and birth outcomes were also similar between the groups, such as induction of labour, pain relief, perineal injury, blood loss and length of stay.

### DISCUSSION

The aim of this study was to evaluate a model of gANC for Somali-speaking women in Sweden. No differences regarding overall ratings of ANC were detected; however, the reduction in mean EPDS score was greater in the intervention group when adjusting for differences at baseline. In late pregnancy, women in gANC were happier with received pregnancy and birth information compared with women in standard care.

Participants in gANC rated their ANC experience at least on par with women who received sANC. The decrease in EPDS mean scores was larger in women in gANC than in sANC, adjusted for differences at baseline. The gANC

Table 2 Women's ratings of satisfaction with specific components of antenatal care ('always or mostly happy with care') in relation to care model in late pregnancy and 2 months post partum

| | Late pregnancy | | | Two months post partum | | |
| --- | --- | --- | --- | --- | --- | --- |
| | Group antenatal care | Standard antenatal care | | Group antenatal care | Standard antenatal care | |
| | n (%) | n (%) | P value | n (%) | n (%) | P value |
| Specifically, were you happy with… | | | | | | |
| How the health of the baby was monitored during pregnancy? | 40 (100.0) | 38 (95.0) | 0.990 | 44 (100.0) | 42 (100.0) | Not Applicable (NA) |
| How your own health was monitored during pregnancy? | 40 (100.0) | 37 (94.9) | 0.463 | 44 (100.0) | 40 (95.2) | 0.454 |
| The management of your physical health?* | 40 (100.0) | 35 (89.7) | 0.117 | | | |
| The support you received to care for your own health? (ie, lifestyle factors) | 39 (97.5) | 32 (82.1) | 0.057 | 42 (95.5) | 38 (90.5) | 0.629 |
| The advice you received to manage common pregnancy disorders? | 37 (92.5) | 29 (74.4) | 0.061 | 44 (100.0) | 34 (81.0) | 0.008 |
| Did the antenatal care midwives… | | | | | | |
| Listen to your own concerns? | 40 (100.0) | 38 (100.0) | NA | 44 (100.0) | 40 (95.2) | 0.454 |
| Make your partner feel welcomed and included? | 37 (92.5) | 34 (97.1) | 1.000 | 40 (93.0) | 39 (92.9) | 1.000 |
| Show respect?* | 39 (97.5) | 37 (94.9) | 0.982 | | | |
| Make you feel welcomed?* | 39 (97.5) | 39 (100.0) | 1.000 | | | |
| Make you feel negatively discriminated against (because of ethnicity, religion, culture, language, etc)?* | 0 (0.0) | 0 (0.0) | | | | |
| Have enough time for you during your visits?* | 40 (100.0) | 39 (100.0) | NA | | | |
| Take your worries seriously?* | 40 (100.0) | 34 (91.9) | 0.212 | | | |
| Keep you informed about the progress of your pregnancy?* | 40 (100.0) | 38 (97.4) | 0.990 | | | |
| Been encouraging and supportive?* | 40 (100.0) | 32 (82.1) | 0.016 | | | |
| Did you feel comfortable to ask questions?* | 40 (100.0) | 39 (100.0) | NA | | | |
| Did you understand the information that you received from the midwives?[1] | 40 (100.0) | 38 (100.0) | NA | | | |
| Do you think that the information you received from the midwives was relevant?* | 39 (100.0) | 37 (92.5) | 0.248 | | | |

*Only in questionnaire 1.

group had relatively high EPDS median scores,[10] when entering the intervention, and higher than the control group,[6] indicating a probable selection bias. It is possible that women who were feeling vulnerable and wanting more support and information from ANC were more interested in participation in gANC. Women who participated in gANC had comparatively low levels of education, no previous births in Sweden, were less involved in working life and perhaps less knowledgeable about Swedish healthcare. Whether or not the larger reduction in mean EPDS scores among women in gANC can be attributed to the intervention or to other aspects of care given the study design, the reduced sample size and the above discussion is uncertain. In a small Norwegian study on postpartum depression among Somali women, the mean EPDS score was 2.97 (SD 3.31), which is very low and might reflect a different understanding of the EPDS questions[69] or a selection of women other than in the present study. In our case, the EPDS questions were asked by the research assistant rather than women completing the EPDS themselves which may have impacted the responses, but it did allow for clarification of items, if they were not clear. Information on history of mental illness or medication was not collected. A still unpublished Swedish study suggests that the Somali version of the EPDS has some major lexical problems. During think-aloud interviews, some Somali-speaking women left items unanswered when responding to the EPDS because they did not understand the meaning of them (oral communication Schytt *et al*), which may also explain some of the extensive missing rates in other surveys using the scale. Challenges with interpretation in research on mental ill health and cultural aspects of mental health also need to be considered.[70] The EPDS has been translated into Somali but is not yet validated,[62] so in the absence of validation, caution is required when interpreting EPDS for this group. Additional or other outcome measures than the EPDS might be appropriate in future trials to evaluate the effects of gANC on Somali women's emotional well-being.

The most interesting differences detected between the gANC and sANC groups were related to secondary

**Table 3** Sufficient information and social support comparing group antenatal care with standard antenatal care

| | Late pregnancy | | | Two months post partum | | |
|---|---|---|---|---|---|---|
| | Group antenatal care | Standard antenatal care | | Group antenatal care | Standard antenatal care | |
| | n (%) | n (%) | P value | n (%) | n (%) | P value |
| **Sufficient information** | | | | | | |
| Ratings for having received enough information about (yes vs no) | | | | | | |
| Why it is important to take iron tablets | 39 (100.0) | 38 (95.0) | 0.368 | 43 (97.7) | 39 (92.9) | 0.576 |
| When to go to hospital for labour | 39 (100.0) | 27 (67.5) | <0.001 | 44 (100.0) | 37 (90.2) | 0.107 |
| Management of female genital mutilation/cutting (FGM/C) | 31 (79.5) | 16 (40.0) | 0.001 | 39 (88.6) | 21 (51.2) | <0.001 |
| Management of a delivery | 36 (94.7) | 17 (42.5) | <0.001 | 44 (100.0) | 27 (64.3) | <0.001 |
| Pain relief during labour | 38 (97.4) | 12 (30.0) | <0.001 | 44 (100.0) | 29 (69.0) | <0.001 |
| Induction of labour | 33 (84.6) | 11 (27.5) | <0.001 | 42 (95.5) | 25 (59.5) | <0.001 |
| Vacuum extraction | 33 (84.6) | 6 (15.0) | <0.001 | 41 (93.2) | 19 (45.2) | <0.001 |
| Caesarean section | 37 (94.9) | 7 (17.5) | <0.001 | 43 (97.7) | 22 (52.4) | <0.001 |
| Father's/partner's role during labour | 35 (94.6) | 17 (42.5) | <0.001 | 43 (97.7) | 26 (61.9) | <0.001 |
| Care at the neonatal care unit | 38 (97.4) | 21 (52.5) | <0.001 | 43 (97.7) | 31 (73.8) | 0.004 |
| Breast feeding | 38 (97.4) | 23 (57.5) | <0.001 | 44 (100.0) | 33 (78.6) | 0.004 |
| Care of the newborn baby | 38 (60.3) | 21 (28.8) | <0.001 | 42 (97.7) | 34 (81.0) | 0.031 |
| Care for you after labour/who to contact after discharge | 37 (94.9) | 15 (37.5) | <0.001 | 44 (100.0) | 29 (72.5) | 0.001 |
| Physical changes during pregnancy* | 39 (100.0) | 30 (75.0) | 0.004 | | | |
| Emotional changes during pregnancy* | 37 (94.9) | 29 (72.5) | 0.001 | | | |
| Taking care of your own health during pregnancy* | 39 (100.0) | 35 (87.5) | 0.074 | | | |
| When to contact the midwife or hospital between the scheduled antenatal care visits* | 39 (100.0) | 32 (80.0) | 0.013 | | | |
| Physical changes after pregnancy† | | | | 44 (100.0) | 29 (69.0) | <0.001 |
| Emotional changes after pregnancy† | | | | 42 (95.5) | 24 (57.1) | <0.001 |
| Taking care of your own health after pregnancy† | | | | 43 (97.7) | 28 (66.7) | <0.001 |
| Do (did) you feel well prepared for labour and birth? | 39 (100.0) | 36 (90.0) | 0.128 | 41 (97.6) | 34 (85.0) | 0.099 |
| **Social support** | | | | | | |
| Do you have someone | | | | | | |
| To socialise with? | 38 (97.4) | 36 (100.0) | 1.000 | 42 (100.0) | 41 (100.0) | Not applicable (NA) |
| To talk with about your problems? | 38 (97.4) | 37 (100.0) | 1.000 | 42 (100.0) | 41 (100.0) | NA |
| To help you if you were sick and needed to be in bed? | 38 (97.4) | 37 (100.0) | 1.000 | 42 (100.0) | 41 (100.0) | NA |
| Who can lend you 500 SEK if you have a sudden need? | 38 (97.4) | 35 (94.6) | 0.963 | 42 (100.0) | 41 (100.0) | NA |
| Who can take care of your children for a while if needed? | 37 (97.4) | 29 (96.7) | 1.000 | 42 (100.0) | 38 (97.4) | 0.970 |
| Who can help you with a temporary place to live next year if you should need it? | 31 (79.5) | 28 (75.7) | 0.902 | 40 (95.2) | 18 (43.9) | <0.001 |
| Have you made new friends through the antenatal care clinic during your pregnancy?† | | | | 41 (97.6) | 10 (25.0) | <0.001 |
| Have you felt lonely and isolated since you had the baby?† | | | | 1 (2.4) | 2 (4.9) | 0.983 |

*Only in questionnaire 1.
†Only in questionnaire 2.
SEK, Swedish kronor.

outcomes. Women attending gANC were more likely to say that they had received sufficient information than women in sANC on most aspects of childbirth studied. Findings on the women's knowledge of danger signs were also promising. A higher proportion of women in gANC gained a better understanding of which healthcare facility to turn to for danger signs like vaginal bleeding and leakage of amniotic fluid. Information and education have been identified as essential in midwifery-led ANC, for adequate knowledge and for women to better understand the healthcare system, and women prefer health professionals who combine clinical knowledge

**Table 4** Obstetric outcomes in relation to care model

| | Group antenatal care n=63 | | Standard care n=73 | | |
| --- | --- | --- | --- | --- | --- |
| | n (%) | Median (IQR) | n (%) | Median (IQR) | P value |
| **Antenatal care** | | | | | |
| Number of antenatal care visits*† (median, IQR) | | 9.5 (8.00–11.00) | | 9.0 (7.25–10.00) | 0.022 |
| ≤6 | 3 (5.6) | | 2 (3.0) | | 0.018 |
| 7–11 | 40 (74.1) | | 61 (92.4) | | |
| >11 | 11 (20.4) | | 3 (4.5) | | |
| Number of visits to specialist care (median, IQR) | | 2.00 (1.00–5.00) | | 1.00 (0.00–3.00) | 0.014 |
| Number of women referred to a specialist (obstetrician) | 49 (90.7) | | 51 (77.3) | | 0.085 |
| Screened for experience of violence‡ | 37 (88.1) | | 37 (84.1) | | 0.822 |
| Attendance at parent education§ | 9 (27.3) | | 3 (8.3) | | 0.056 |
| **Health parameters** | | | | | |
| Haemoglobin (lowest value) | | 111 (106–118) | | 108 (104–114) | 0.181 |
| Haemoglobin (last value prior to birth) | | 122 (118–128) | | 120 (115–126) | 0.218 |
| S-ferritin (lowest value) | | 24 (16–54) | | 23 (13–42) | 0.539 |
| Weight gain during pregnancy (kg) | | 8.0 (4.3–12.0) | | 7.0 (4.0–12.0) | 0.589 |
| Gestational diabetes mellitus | 7 (11.1) | | 7 (9.6) | | 0.771 |
| **Labour and birth** | | | | | |
| Induction of labour | 8 (17.8) | | 10 (16.4) | | 1.000 |
| Oxytocin for dystocia | 16 (42.1) | | 23 (37.7) | | 0.823 |
| Pain relief | | | | | |
| Epidural | 9 (14.5) | | 5 (7.4) | | 0.259 |
| Gas (nitrous oxide) | 33 (53.2) | | 35 (50.7) | | 0.912 |
| Other (TENS (Transcutaneous electrical nerve stimulation), sterile water injection, bath, local or regional anesthesia) | 1 (1.6) | | 2 (2.9) | | 1.000 |
| Mode of birth | | | | | |
| Vaginal | 42 (77.8) | | 52 (78.8) | | 0.064 |
| Assisted vaginal delivery | 0 (0.0) | | 6 (9.1) | | |
| Elective caesarean | 8 (14.8) | | 4 (6.1) | | |
| Emergency caesarean | 4 (7.4) | | 4 (6.1) | | |
| Perineal injury (degree III–IV) | 0 (0.0) | | 4 (5.8) | | 0.121 |
| Blood loss (within 2 hours) (mL) | | 480 (300–650) | | 400 (300–538) | 0.345 |
| Breast feeding on delivery ward | 50 (98.0) | | 61 (96.8) | | 1.000 |
| Length of stay (in hospital after birth) | | 1.00 (0.0–2.0) | | 1.00 (0.0–2.3) | 0.590 |
| Discharged the same day as giving birth | 23 (43.4) | | 30 (46.9) | | 0.850 |
| **Infant outcomes** | | | | | |
| Live births¶ | 62 | | 73 | | |
| Stillborn | 1 (1.6) | | 1 (1.4) | | 1.000 |
| Gestational age (weeks) | | 40.0 (39.0–40.0) | | 40.0 (39.0–40.3) | 0.989 |
| Birth weight (g) | | 3580 (3180–3778) | | 3490 (3166–3776) | 0.839 |
| Small for gestational age | 7 (11.1) | | 0 (0.0) | | 0.011 |
| Large for gestational age | 3 (4.7) | | 2 (2.7) | | 0.532 |
| Apgar score <7 at 5 min | 0 (0.0) | | 1 (1.7) | | 1.000 |

Continued

**Table 4** Continued

| | Group antenatal care n=63 | | Standard care n=73 | | |
|---|---|---|---|---|---|
| | n (%) | Median (IQR) | n (%) | Median (IQR) | P value |
| Umbilical cord pH (arterial) (median, IQR) | | 7.24 (7.20–7.31) | | 7.26 (7.23–7.31) | 0.224 |
| Umbilical cord pH (vein) (median, IQR) | | 7.35 (7.29–7.37) | | 7.35 (7.30–7.37) | 0.795 |
| Neonatal intensive care | 1 (1.6) | | 5 (7.2) | | 0.212 |
| Postnatal check-up | | | | | |
| Postnatal visit | 51 (81.0) | | 47 (64.4) | | 0.032 |
| Fully breastfeeding at 4 weeks after birth | 25 (39.7) | | 24 (32.9) | | 0.410 |
| Body mass index (kg/m$^2$) at postnatal check-up visit | | 28.34 (25.56–31.64) | | 29.63 (26.44–33.12) | 0.379 |

*Not adjusted for gestational age.
†The recommended number of prenatal visits in Sweden is nine for a normal 40-week pregnancy.
‡Screening for violence is recommended in the Swedish National Guidelines for antenatal care.
§Other than group antenatal care.
¶One twin birth in standard care.

and skills with interpersonal and cultural competence.[71] In this study, as in others, Somali-born women had a high prevalence of risk factors during pregnancy, for example, previous perinatal mortality and obesity. Our findings suggest some potential for gANC to improve information provision and knowledge acquisition among pregnant Somali-born women, especially women residing in Sweden for <10 years.

Additionally, gANC may have played a role in strengthening social networks as intended. Nearly all women in gANC said they had someone who could help with a temporary place to live should they need it, compared with 25.0% of women in standard care 2 months post partum. Additionally, nearly all women in gANC reported that they had made new friends through the ANC clinic during pregnancy, compared with only half of women in standard care.

The women in gANC had more appointments with the midwife compared with women in standard care, possibly because individual appointments were shorter in gANC. As a result, additional appointments were booked, rather than extending the time for each appointment when more time was needed. gANC is a complex intervention[72] and it is therefore not possible, nor appropriate, to determine the separate contributions of different components. The components have been studied together because of their hypothesised capacity jointly to make a difference to the provision of care. A case study from the USA suggests that an underlying mechanism in which group prenatal care is effective is through increased quantity and quality of patient and practitioner time together and improved communication, fostering greater opportunity for cross-cultural exposure and decreasing factors like the clinician's implicit or explicit bias and racism.[73] A process evaluation of the Hooyo Project has assessed the overall feasibility of developing, implementing and testing gANC for Somali-Swedish women, including the perspectives

of the midwives, where the different components of this gANC model have been examined in more depth.[74]

The inconsistent evidence about the impact of gANC on improved ratings of care and obstetric outcomes may also be due to substantial differences between settings in the quality of sANC, as suggested in a Canadian study that found that gANC was comparable with individual care on most outcomes measured.[47] In the USA, however, there are greater disparities in the care received between subgroups, such as for women from minority groups or with low socioeconomic status who are uninsured.[75] To achieve greater prenatal health equity for migrant women, there is a need to continue to investigate specific interventions in ANC focused on pregnant women and their families, on maternity care health professionals and on the system in which they work.

### Strengths and limitations

This was the first controlled evaluation of gANC in Sweden attempting to address the particular needs of migrant women, with a participatory approach. This is of particular importance in groups of women who are often excluded in research, are hard to reach or when studying sensitive health issues.[76] Moreover, this intervention and the evaluation were conducted in a real-world setting. Recruitment and retention of participants proved challenging. The recruitment pace was slower than anticipated and we had fewer eligible women than expected that registered in ANC and could not fill the groups in a satisfactory way. Recruitment had to be terminated before reaching the goal because of these factors in combination with resource limitations. We did not reach the goal of 70 women in each group, reducing the power to detect differences on overall rating of care between the groups. For the EPDS, however, we reached our recruitment goal of 63 women in each group. The two-stage recruitment process with midwives providing initial information and

then subsequent recruitment of those showing interest by a bilingual research assistant may have made it harder to recruit women. Direct initial contact with a Somali-speaking research assistant might have made it easier for women to receive enough information in their own language to agree to participate. While the research assistants were pivotal in recruitment, retention and data collection in this study, women's responses were not given anonymously, which may have affected participation and responses. Additionally, participation in research was likely to be unfamiliar for some of the women.

For this study, women were recruited from one site only, and the possible selection bias of participants recruited to the study further limits the generalisability of the results. We do not know if the results are representative for similar Somali-Swedish groups of women. The findings can however inform future randomised controlled trials and be useful for future power calculations.

## CONCLUSIONS

This study suggests potential for gANC to improve information provision and knowledge acquisition among pregnant Somali-born women with residence in Sweden <10 years. The study found few significant differences between the groups on women's overall ratings of care and emotional well-being measured with EPDS. The study demonstrated feasibility of assessing the included outcomes with Somali-born women. Moreover, the study suggests that a range of outcome measures are important when evaluating language-supported gANC, and that a validation of the Somali version of the EPDS is needed. Finally, an adequately powered randomised trial would be needed for the effectiveness of gANC to be robustly assessed.

**Acknowledgements** We are grateful to all the women and midwives who participated and shared their experiences at different stages of this project, and to the Reference Group. We wish to commemorate midwife Ros-Mari Casselbrant who passed away in October 2019. We wish to acknowledge the following contributors specifically: research assistants Fardosa Hassen Ahmed and Aisha Adan, interpreter Hamdi Sheek Hussein and David Grannas, statistician at Biostatistics Core Facility, Karolinska Institutet.

**Contributors** All authors (MA, UB, ES, EA, RS, BE) were part of the research team that developed the intervention and the questionnaires. UB and MA monitored the day-to-day implementation of the intervention and data collection at the clinic. MA and ES were responsible for the analyses. All authors (MA, UB, ES, EA, RS, BE) have read, commented on and approved the final manuscript. Author ES is acting as guarantor.

**Funding** This work was supported by the Swedish Research Council (grant number 2015-02470), Forte (grant number 2016-00957) and the Doctoral School in Health Care Sciences, Karolinska Institutet (grant number N/A).

**Competing interests** None declared.

**Patient and public involvement** Patients and/or the public were involved in the design, or conduct, or reporting, or dissemination plans of this research. Refer to the Methods section for further details.

**Patient consent for publication** Not required.

**Ethics approval** This study involves human participants and ethical approval was obtained from Stockholm Ethical Review Board (reference number 2015/1703-31/1). Participants gave informed consent to participate in the study before taking part.

**Provenance and peer review** Not commissioned; externally peer reviewed.

**Data availability statement** Data are available upon reasonable request. Deidentified participant data are available upon reasonable request from the corresponding author: Malin Ahrne (malin.ahrne@ki.se).

**ORCID iDs**
Malin Ahrne http://orcid.org/0000-0002-3946-5847
Ulrika Byrskog http://orcid.org/0000-0002-1713-6014
Erica Schytt http://orcid.org/0000-0002-6018-9082

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
