## [Reviewer comments · BMJ Open]

ARTICLE DETAILS

TITLE (PROVISIONAL)	Group antenatal care compared with standard antenatal care for Somali-Swedish women: a historically controlled evaluation of the Hooyo-project
AUTHORS	Ahrne, Malin; Byrskog, Ulrika; Essen, Birgitta; Andersson, Ewa; Small, Rhonda; Schytt, Erica

VERSION 1 – REVIEW

REVIEWER	Kearney, Lauren The University of Queensland, School of Nursing, Midwifery and Social Work
REVIEW RETURNED	08-Aug-2022

GENERAL COMMENTS	Thank you for the opportunity to review the manuscript entitled “Comparing group antenatal care and standard antenatal care for Somali-Swedish women. Findings from the Hooyo project- a historically controlled evaluation”, which aimed to compare language-supported group antenatal care (gANC) and standard antenatal care (sANC) for Somali-born women in Sweden, measuring overall ratings of care and emotional wellbeing, and testing the feasibility of the outcome measures. Abstract: • Clear and well written, no changes Introduction: • First sentence is ambiguous – while I understand this relates to global health priorities – this should be clearer here• Line 27 – should read ‘general population of migrant...’• Write SGA and other abbreviations out in full in first instance, and review to check you need to abbreviate, as may only be used 1 or 2 times, in which case avoid abbreviations Patient and public involvement: • I would like to commend the authors on the rigour applied to this component of the study. It is an area often overlooked, but clearly this area was well considered and applied – well done! Discussion: • Suggest moving the last sentence, directly prior to strengths and limitations, into the paragraph above, as it’s on it’s own as an ‘orphan’ sentence Strengths and limitations: • Authors need to acknowledge the limitation of selection bias here, as this also mitigates the generalisability of the results (which they have vaguely mentioned, but should be explicit here) The authors are to be congratulated on a very important area of study. Well done.
---

REVIEWER	Dadras, Omid
-----------------	--------------

	Walailak University
REVIEW RETURNED	08-Nov-2022

GENERAL COMMENTS	This manuscript compared language-supported group antenatal care (gANC) and standard antenatal care (sANC) for Somali-born women (>25 months pregnant) in Sweden and reported on the overall care and emotional well-being of included women. The authors recruited two groups of Somali women at an ANC clinic in Sweden and collected the data at three points during the intervention. The results suggested an improvement in the overall well-being and pregnancy experience among the women. The authors followed a sound and rigorous protocol and managed to produce impressive results that could inform the future policy and interventions for immigrant mothers' health and I'd be happy to recommend this manuscript for publication. However, there few points that should be addressed before as below: 1- I assume the authors conducted a quasi-experimental trial as the randomization component is missing in the participant recruitment but they haven't mentioned this anywhere. 2- Although the authors recruited the participants in a non-random manner, there are other components that should be considered in conducting a trial to improve the accuracy and reduce the bias in collected data. I assume that the authors have considered them but probably missed mentioning them in the manuscript. The matter of blinding those who collect the data is the main concern. The interviewers are assumed to be masked to the assignment of the participants in the intervention and control groups. 3- The other bias that may distort the estimates is the drop-out and the characteristics and the intention of those who fail or reject to continue the intervention. 4- There are several missing data at the baseline characteristics such as education or language, particularly on the control group side, and I'm wondering why it happened when the patient's record is available moreover, adjusting for these baseline characteristics is the best way to obtain accurate estimates and account for the sociodemographic inequalities between intervention and control group. The question also stands for how the author handles this missing data.
--

VERSION 1 – AUTHOR RESPONSE

Reviewer: 1

Introduction:

• **First sentence is ambiguous – while I understand this relates to global health priorities – this should be clearer here**

The sentence has been revised.

• **Line 27 – should read ‘general population of migrant...’**

The grammar has been corrected.

• **Write SGA and other abbreviations out in full in first instance, and review to check you need to abbreviate, as may only be used 1 or 2 times, in which case avoid abbreviations**

Abbreviations have been checked. BMI, OR and CI have now been written out in full. The abbreviations that are used now are used more than twice.

Discussion:

- **Suggest moving the last sentence, directly prior to strengths and limitations, into the paragraph above, as it's on it's own as an 'orphan' sentence**

Thank you, this has been done.

Strengths and limitations:

- **Authors need to acknowledge the limitation of selection bias here, as this also mitigates the generalisability of the results (which they have vaguely mentioned, but should be explicit here)**

The limitation of selection bias has been more explicitly expressed.

Reviewer: 2

- 1- I assume the authors conducted a quasi-experimental trial as the randomization component is missing in the participant recruitment but they haven't mentioned this anywhere.**

We have clarified this under Study design, and changed the wording from "controlled evaluation" to "quasi-experimental trial".

- 2- Although the authors recruited the participants in a non-random manner, there are other components that should be considered in conducting a trial to improve the accuracy and reduce the bias in collected data. I assume that the authors have considered them but probably missed mentioning them in the manuscript. The matter of blinding those who collect the data is the main concern. The interviewers are assumed to be masked to the assignment of the participants in the intervention and control groups.**

With this design it was not possible to blind the intervention, neither to the participants, caregivers nor to the research team.

- 3- The other bias that may distort the estimates is the drop-out and the characteristics and the intention of those who fail or reject to continue the intervention.**

The text related to bias has been developed in the Strengths and limitations section. The dropouts have been analysed separately in terms of EPDS scores, which is described in the paper.

- 4- There are several missing data at the baseline characteristics such as education or language, particularly on the control group side, and I'm wondering why it happened when the patient's record is available moreover, adjusting for these baseline characteristics is the best way to obtain accurate estimates and account for the sociodemographic inequalities between intervention and control group. The question also stands for how the author handles this missing data.**

The data on education and language was self-reported and collected from the questionnaires, and such information is not included in the patient records.

The issue of adjusting for baseline characteristics is a valid point. The reasons for not adjusting for baseline characteristics is that this was mainly a descriptive feasibility study, not randomized and with limited power to detect effectiveness of the intervention so we believed it was a strength in this small study not to adjust, but rather to present the data descriptively. The control group and the intervention group were also largely similar, with few differences. Missing data is a weakness in this small study, but imputation was further not assessed as suitable.

*** **

VERSION 2 – REVIEW

REVIEWER	Kearney, Lauren The University of Queensland, School of Nursing, Midwifery and Social Work
REVIEW RETURNED	05-Jan-2023

GENERAL COMMENTS	The feedback has been addressed well. This manuscript provides important evidence to support innovation and flexibility in models of care for vulnerable women.
---

REVIEWER	Dadras, Omid Walailak University
REVIEW RETURNED	20-Dec-2022

GENERAL COMMENTS	All my comments from the first review have been addressed in the revised manuscript and I'm glad to recommend the publication of this manuscript.
---